Contrast based circular approximation for accurate and robust optic disc segmentation in retinal images

Sigut Jose sigut@isaatc.ull.es 1
Nunez Omar omar@isaatc.ull.es 1
Fumero Francisco 1
Gonzalez Marta 2
Arnay Rafael 1
1 Department of Computer Engineering and Systems, Universidad de La Laguna , San Cristobal de La Laguna , Spain
2 Department of Ophthalmology, Hospital Universitario de Canarias , San Cristobal de La Laguna , Spain
Kalpathy-Cramer Jayashree
Electronic publication date: 2017 Sep 7
Publication date: 2017
Volume: 5
Electronic Location ID: e3763
Received 2017 Jun 2; Accepted 2017 Aug 15
Copyright: ©2017 Sigut et al.
Copyright year: 2017
Copyright holder: Sigut et al.
License: This is an open access article distributed under the terms of the Creative Commons Attribution License, which permits unrestricted use, distribution, reproduction and adaptation in any medium and for any purpose provided that it is properly attributed. For attribution, the original author(s), title, publication source (PeerJ) and either DOI or URL of the article must be cited.
License URL: https://creativecommons.org/licenses/by/4.0/

Keywords: Optic disc, Retinal image analysis, Image analysis, Optic disc localization, Optic disc segmentation, Glaucoma detection

Funding: The authors received no funding for this work.

==============================
A new method for automatic optic disc localization and segmentation is presented. The localization procedure combines vascular and brightness information to provide the best estimate of the optic disc center which is the starting point for the segmentation algorithm. A detection rate of 99.58% and 100% was achieved for the Messidor and ONHSD databases, respectively. A simple circular approximation to the optic disc boundary is proposed based on the maximum average contrast between the inner and outer ring of a circle centered on the estimated location. An average overlap coefficient of 0.890 and 0.865 was achieved for the same datasets, outperforming other state of the art methods. The results obtained confirm the advantages of using a simple circular model under non-ideal conditions as opposed to more complex deformable models.

Introduction

The optic disc (OD) is one of the main anatomical structures in retinal images. For that reason, its location and segmentation are very important tasks in retinal image analysis and processing. OD analysis is used in the diagnosis of several retinal diseases, most importantly, glaucoma (Haleem et al., 2013; Almazroa et al., 2015). Moreover, it can be used as a landmark for other retinal features such as the fovea or even as a starting point for vessel tracking.

Localization and segmentation are usually considered as two separate tasks. OD localization is commonly defined as the task of determining the coordinates of a pixel belonging to the optic disc, usually the center. OD segmentation consists of delimitating the disc boundary.

The OD usually appears in retinal images as a bright yellowish region, approximately circular in shape and with blood vessels converging towards its center. These well-defined features may give the impression that locating and segmenting the OD is an easy problem. However, very low contrast images and retinal pathologies such as exudates and different types of lesions turn the problem into very challenging as it is evident by the performance results found in recent publications where the algorithms are tested on large databases.

Many different methods have been proposed in the last years for OD localization and most of them try to take advantage of the basic features mentioned above, the expected appearance of the disc in retinal images and the high density of vessels in the region of interest. Aquino, Gegúndez-Arias & Marin (2010) use three independent detection methods on the green channel to provide three different candidate pixels and a voting procedure is applied to select the best one. In Lu (2011), a circular transformation is designed to simultaneously capture the circular shape of the OD and the image variation across the OD boundary. The center of the OD is determined by the pixels with the maximum variation along multiple radial segments. Giachetti, Ballerini & Trucco (2014) use vascular and brightness priors based on simple probabilistic detectors, one related to radial symmetry and the other related to vessel density. The work by Mendonça et al. (2013) presents a new technique based on the entropy of vascular directions to assess the convergence of vessels around an image point. Abdullah, Fraz & Barman (2016) use the Circular Hough Transform (CHT) to search for the OD center.

With respect to OD segmentation, most existing methods can be roughly categorized as template-based, deformable model-based, morphology based and, more recently, machine learning methods. In practice, a combination of them is the usual approach. Template-based techniques are mainly based on the assumption of circularity or ellipticity of the OD. Zhu & Rangayyan (2008), Lu (2011) and Aquino, Gegúndez-Arias & Marin (2010) use the CHT to approximate the OD boundary. Moreover, Aquino, Gegúndez-Arias & Marin (2010) provide an interesting discussion about the convenience of choosing a simple circular approximation as opposed to more complex models. Zheng et al. (2013) and Giachetti, Ballerini & Trucco (2014) prefer an elliptical approximation. In the work by Giachetti et al., the OD boundary is determined using refined elliptic contours which are finally improved with snake-based algorithms. Regarding the deformable model approaches, Lowell et al. (2004) extract the OD boundary using a global elliptical parametric model with a local deformation. Xu et al. (2007) propose a deformable model which combines active contour deformation with clustering. Yu et al. (2012) use a fast and hybrid level set model with optimized parameters. Mary et al. (2015) propose an OD segmentation scheme by means of the gradient vector flow model. Although these methods can, in theory, provide accurate OD segmentations, they are also very sensitive to artifacts which may be present in the retinal image. Morphology-based techniques primarily make use of the brightness and shape properties of the OD. Reza, Eswaran & Hati (2009) use morphological opening, extended maxima operator, and watershed transformation with minima imposition to segment the OD. Welfer et al. (2010) propose a new adaptive morphological method to detect the optic disc center and the optic disc rim. Morales et al. (2013) combine mathematical morphology with principal component analysis. Marin et al. (2015) present a methodology which performs a set of iterative opening–closing morphological operations on an intensity image and the result is thresholded taking also into account blood vessel confluence. The OD region is finally obtained by applying the CHT on the output of a Prewitt edge detector. More recently, machine learning methods for OD segmentation have become popular as they provide a powerful tool for feature classification using learned models. Abramoff et al. (2007) perform pixel classification on stereo pairs using an optimal subset of 12 features. To overcome the limitation of classification at the pixel level, Cheng et al. (2013) propose a superpixel based method to segment the OD and the optic cup. Lim et al. (2015) provide a solution based on convolutional neural networks (CNN). Roychowdhury et al. (2016) propose a three-step classification based OD segmentation algorithm. Zilly, Buhmann & Mahapatra (2017) present a novel method using ensemble learning based also on CNN architectures.

In general, the analysis of the literature reveals that most existing methods do not easily handle the problem of the variable appearance of the optic disc in retinal images under non ideal conditions. Model based approaches face the challenge of reliably representing the OD contour and its associated features in the presence of anomalies. Morphology based methods have difficulties in dealing with varying image color, unpredictable intersections of vessels with the OD region and effects of different pathologies. Learning based approaches suffer from the drawbacks inherent to building the appropriate training set. For this reason, a new method for OD localization and segmentation is presented.

The localization procedure combines vascular and brightness information to provide the estimation of the optic disc center which is the starting point for the segmentation task. The segmentation strategy is partly inspired by the work of Aquino, Gegúndez-Arias & Marin (2010) who show that a simple circular model can provide lower overlapping error and better bias–variance trade-off than complex deformable shape/appearance models. As mentioned above, they use the Circular Hough Transform to provide the best approximation to the OD boundary. The approach followed in this work is based on the idea that the average contrast between the outer and inner rings of a circular approximation to the OD boundary is higher than in other parts of the region of interest. By doing so, a much better performance has been obtained as it will be shown in the ‘Results’ section.

In our opinion, the main contribution of this paper is twofold. First, experimental results show that the simple solution proposed for OD segmentation can outperform other state of the art methods, in particular, in terms of overlap rates. Second, the proposed segmentation method can be used in conjunction with any localization procedure, other than the one suggested in this work, which could even lead to an improvement in the results according to the performance obtained for manual localization.

The rest of the paper is organized as follows. In the ‘Materials and Methods’, the proposed methodology for OD localization and segmentation is described together with the retinal image databases and the measures which have been used for evaluating the performance. The experimental results which were obtained are presented and compared with other methods in the ‘Results’. The paper concludes with the ‘Discussion and Conclusion’ section.

Materials & Methods

Materials and performance evaluation

Two publicly available databases of retinal images have been considered to evaluate the performance of the proposed localization and segmentation methods. The main reason for this choice is the widespread use of these datasets and also the availability of a ground truth which can be used in the same conditions for any researcher. There exist other popular databases but the ground truth is not always accessible.

The Messidor database (Decencière et al., 2014) was created in the framework of diabetic retinopathy screening and diagnosis. It consists of 1,200 images acquired in three different ophthalmology departments using a 3CCD color video camera on a Topcon TRC NW6 non-mydriatic retinograph with a 45°  FOV and three different resolutions: 1,440 × 960, 2,240 × 1,488 and 2,304 × 1,536 pixels. This dataset has been used in most of the recent works mainly because of its size, much bigger than other public datasets, and due to the availability of a ground truth with contours traced by an expert from the University of Huelva (2012).

The ONHSD database (Lowell et al., 2004) consists of 99 fundus images with a resolution of 760 × 570 pixels taken from 50 patients in the context of a diabetic retinopathy screening program. The images were captured using a Canon CR6 45MNf camera with a 45°  FOV. As suggested by Lowell et al. (2004), only a subset of 90 images is considered due to the very bad quality of the discarded images. A ground truth is also provided.

In order to evaluate the performance of the proposed methods, different measures are proposed. The OD localization is usually considered as successful if the estimated center lies within the boundary of the corresponding ground truth and that is the approach followed in this work. The error in the localization is evaluated in terms of the error distance, Dcexp,creal, between the real location and the experimental one. Since the size of the retinal images under consideration may be different, a normalized error distance, D∗cexp,creal, becomes a more convenient measure. (1) D∗c exp,creal=Dcexp,crealR

where R is the radius of a circle with an area equal to the size of the corresponding ground truth mask.

For the purpose of evaluating the performance of the segmentation method, the Jaccard and Dice coefficients as well as the mean average distance between the true and estimated OD boundaries have been chosen.

The Jaccard and Dice coefficients are measures of the similarity between two sets. The Jaccard coefficient, JC, is defined as the ratio between the intersection and union of the automatic segmentation of the OD, SOD, and the corresponding ground truth, STruth. (2) JCSOD,STruth=SOD∩STruthSOD∪STruth.

The Dice coefficient, DC, is defined as twice the ratio between the intersection of SOD and STruth, and the sum of their sizes. (3) DCSOD,STruth=2SOD∩STruthSOD+STruth.

As opposed to the JC and DC coefficients which are intended for evaluating the degree of overlap between regions, the mean average distance, MAD, is used to measure the distances between the closest points on their boundaries. Let us denote by BOD=a1,a2,…,aN and BTruth=b1,b2,…,bM, the finite sets of points belonging to the boundaries of the segmented OD and the corresponding ground truth, respectively. The MAD between the two sets is defined as follows: (4) MADBOD,BTruth=12∑i=1Ndai,BTruthN+∑j=1Mdbj,BODM

where dai,BTruth=jminbj−ai and dbj,BOD=iminai−bj.

It is important to mention that the parameters of the methods for OD localization and segmentation have been experimentally determined using 40 images from the ONHSD dataset. The performance evaluation has been carried out using the whole ONHSD dataset and the Messidor database.

Method for optic disc localization

The method for the estimation of the optic disc center consists of two main steps: creating a mask based on vascular information to shrink the search space, and filtering the image with a detector which combines vascular and brightness information. The only preprocessing which was found to have a significant impact on the results consisted of resizing the original image using cubic interpolation so that the diameter of the retina becomes equal to 540 pixels as in Marin et al. (2015). By doing so, the proposed methodology can be used without the necessity of adapting the parameters of the algorithms. In the case of using a different angle for capturing the images, the parameters should be readjusted accordingly.

Shrinking the search space using vessel information

The first step exploits the fact that the OD is the entry point for the major blood vessels that supply the retina. Consequently, the OD region presents usually a high vessel density and can be seen also as a convergence point of this structure. The complementary of the green channel in the RGB color space, Gc, is taken as the input image for this stage since it is known to provide a good vessel-background contrast. Vessel enhancement, VE, is carried out by applying the top-hat operator T8 to Gc using a disc of 8 pixels in radius as the structuring element. (5) VE=T8Gc.

Vessel density, VD, is calculated as the difference between the output of two averaging filters on VE, an averaging filter of size 80 × 40 and an averaging filter of size 80 × 120. (6) VD=Av80×40VE−Av80∗120VE.

The result is normalized dividing by its maximum. The rationale behind this is not only to capture the high density of vessels in the OD region but also take advantage of the fact that the density on both sides of it is significantly lower. A binarized version, VDb, is obtained by thresholding VD using a threshold of 0.3.

The convergence of the branches of the vascular tree, VC, is estimated by finding the intersections of lines which are used to approximate the branches of this tree. The lines are the result of the application of the Hough Transform (HT) to the output of a Canny edge detector, EDCanny, computed on the vessel-enhanced image, VE, so that the value of each pixel of VC corresponds to the number of Hough line intersections at that point. Due to the noisy nature of this calculation, the resulting image is filtered to obtain an averaged version using a circular averaging filter of radius 40. (7) VC=Avr=40HTEDCannyVE.

A binarized version, VCb, is obtained by thresholding VC using a threshold of 0.3.

The two previous masks are combined using a logical AND operator to provide the final constraint mask, VDCb, based on vascular information. (8) VDCb=logical ANDVDb,VCb.

Figure 1 shows an example of the different operations involved in the calculation of VDCb. It is interesting to remark the role of VCb in the discarding of exudates. Very few lines are detected by the Hough Transform in these unwanted regions as shown in Figs. 1E and 1F.

Figure 1 Calculation of the mask based on vessel information.

(A) Original image, (B) Vessel enhanced image, (C) Vessel density, (D) Vessel density mask, (E) Hough Transform of the Canny edge detector output, (F) Vessel convergence, (G) Vessel convergence mask, (H) Logical AND of vessel density and vessel convergence masks.

Building the optic disc detector

The intensity image, I, in the second step is calculated as the sum of the three RGB color channels so that the OD usually appears as the brightest region but not always due to artifacts. In fact, the vessel pixels in I which are within the OD may take low values. In order to compensate for this low intensity, a vessel mask, VM, is superimposed on the intensity image and the values of the pixels in the mask are set to the maximum value in I to provide the input image, Ic, for the OD detector. The vessel mask is obtained by thresholding VE to retain only those pixels with high values. (9) VM=maxI,ifVE>percentile98.5VE0,otherwise

(10) Icp= maxIp,VMp

where p is any pixel in the image.

The OD detector, DOD, is finally calculated as the difference between the output of two averaging filters on Ic, a circular averaging filter of radius 40 and an averaging filter of size 80 × 160. (11) DOD=Avr=40Ic−Av80∗160Ic.

The x and y coordinates where DOD reaches its maximum value after the application of the constraint mask VDCb are taken as the best approximation to the optic disc center. Figure 2 shows an example of the different operations involved in this second step.

Figure 2 Operations for OD localization.

(A) Intensity image with superimposed vessel mask, (B) Output of OD detector, (C) Application of the constraint mask based on vessel information to the output of the OD detector, (D) Estimated OD center.

Method for optic disc segmentation

The method for OD segmentation has been implemented on a subimage of the original RGB retinography by cropping it around the estimated center. The size of the cropping window depends on the resolution of the image under consideration and has been chosen to completely contain the OD and include a certain amount of the background around. In this way, a more robust and efficient segmentation can be achieved due to reduced space for search and consequently reduced number of artifacts present in the whole image.

The intensity image I defined in the previous section is taken as the basis for the calculation of the contrast between the regions of interest. For that purpose, a set of concentric circles were considered with center xc,yc, coincident with the center of the window, and increasing radii in increments of 1%, ranging from 25% to 45% of the side length of the cropping window. For each reference radius r, an outer and inner ring are defined as shown in Fig. 3.

Figure 3 Inner and outer ring for a reference radius r.

The outer ring is the region lying between the two circles with the biggest radii and the inner ring is the region lying between the two circles with the smallest radii. The thickness of the ring was set to 4% of the side length of the cropping window. A contrast measure is defined as the subtraction of the average intensity in the outer ring from the average intensity in the inner ring: (12) CMr=avIinnerr−avIouterr.

The circle for which CM attains the maximum value is taken as the best approximation to the OD contour under these assumptions.

We have found that the pure circular approximation obtained in this way may become too rigid in some cases. For this reason, an alternative solution is presented which retains the original simplicity but allowing some more flexibility in the form of considering four fractions of a circle instead as shown in Fig. 4. This is the approach which has been followed for OD segmentation.

Figure 4 Fractions of the circle considered for OD segmentation.

(A) Left and right semicircles with their respective radii, (B) Top and bottom semicircles with their respective radii.

The same procedure as described above is applied to each of the circle sections so that four different radii rumax,rdmax,rlmax,rrmax are computed for which the maximum contrast is reached in each section. The final OD boundary is computed as a circle whose center xcmax,ycmax and radius rmax are calculated as follows: (13) xcmax=2xc+rrmax−rlmax,ycmax=2yc+rdmax−rumax

(14) rmax=rumax+rdmax+rlmax+rrmax4.

Results

Optic disc localization

The method proposed for optic disc localization achieves a 100% success rate in the ONHSD database and a 99.58% success rate (5 failures) in the Messidor database. Figure 5 shows the values of D∗cexp,creal obtained for the Messidor and ONHSD datasets with mean values of 0.089 and 0.099, respectively. Table 1 shows a comparison with other methods for both datasets.

Figure 5 Values of D∗cexp,creal obtained for the Messidor (A) and ONHSD (B) datasets.

Table 1 OD localization accuracy comparison with other methods.

Authors	Database	Accuracy	
Our method	Messidor	99.58%	
	ONHSD	100%	
Abdullah, Fraz & Barman (2016)	Messidor	99.25%	
	ONHSD	100%	
Roychowdhury et al. (2016)	Messidor	100%	
Marin et al. (2015)	Messidor	99.75%	
Giachetti, Ballerini & Trucco (2014)	Messidor	99.83%	
Ramakanth & Babu (2014)	Messidor	99.42%	
Mendonça et al. (2013)	Messidor	99.75%	
Suero et al. (2013)	Messidor	99.25%	
Yu et al. (2012)	Messidor	99.08%	
Sekar & Nagarajan (2012)	Messidor	99.58%	
Lu (2011)	Messidor	99.75%	
Aquino, Gegúndez-Arias & Marin (2010)	Messidor	98.83%	

Figures 6 and 7 show some examples of OD localization in the Messidor and ONHSD datasets.

Figure 6 OD localization in images from Messidor database.

Figure 7 OD localization in images from ONHSD database.

Optic disc segmentation

Table 2 shows the performance of our method for optic disc segmentation in terms of the measures described in the materials and performance evaluation section. As it can be seen, two different scenarios have been considered, a completely automatic procedure where the OD center is estimated by means of the localization method explained in this paper and another ideal situation in which the OD center is obtained from the provided ground truth, i.e., manual localization. We have done so to remark the influence of OD localization in the proposed segmentation method and to set a bound on the maximum reachable performance. The average computation time is 1.28 s per image for the automatic version and 0.07 s. for the manual version so it becomes clear that most of the time is consumed in the localization of the OD. A comparison with other methods has also been included in the table. It should be noted that not all the measures are available for the different methods considered.

The percentages of images for different intervals of the overlapping coefficient JC are shown in Table 3 as well as a comparison with other methods for the Messidor dataset due to the unavailability of data for ONHSD.

Figure 8 shows some examples of correct and incorrect segmentations in the Messidor dataset and Fig. 9 shows some examples of correct and incorrect segmentations in the ONHSD dataset using the proposed method.

Table 2 Performance comparison of OD segmentation methods.

Methods	JC¯	DC¯	MAD¯	
Messidor				
Our method (manual localization)	0.915	0.954	3.809	
Our method (automatic localization)	0.890	0.939	5.163	
Zilly, Buhmann & Mahapatra (2017)	0.90	–	–	
Abdullah, Fraz & Barman (2016)	0.879	0.934	–	
Roychowdhury et al. (2016)	0.837		3.9	
Dashtbozorg, Mendonça & Campilho (2015)	0.886	0.937	3.160	
Lim et al. (2015)	0.888	–	–	
Marin et al. (2015)	0.87	0.92	6.17	
Giachetti, Ballerini & Trucco (2014)	0.879	–	–	
Cheng et al. (2013)	0.875	–	–	
Morales et al. (2013)	0.8228	0.8950	4.0759	
Yu et al. (2012)	0.844	–	–	
Aquino, Gegúndez-Arias & Marin (2010)	0.86	–	–	
ONHSD				
Our method (manual localization)	0.893	0.942	2.225	
Our method (automatic localization)	0.865	0.924	2.914	
Abdullah, Fraz & Barman (2016)	0.851	0.910	–	
Dashtbozorg, Mendonça & Campilho (2015)	0.834	0.917	2.422	
Morales et al. (2013)	0.804	0.887	3.275	

Table 3 Percentages of images for intervals of JC values.

Methods	JC¯	JC ≥ 0.95	JC ≥ 0.90	JC ≥ 0.85	JC ≥ 0.80	JC ≥ 0.75	JC ≥ 0.70	
Messidor								
Our method (manual localization)	0.915	23%	75%	91%	96%	98%	99%	
Our method (automatic localization)	0.890	12%	61%	84%	92%	95%	97%	
Roychowdhury et al. (2016)	0.837	–	20%	48%	–	83%	97%	
Dashtbozorg, Mendonça & Campilho (2015)	0.886	23%	66%	81%	87%	92%	94%	
Lim et al. (2015)	0.888	7%	61%	86%	92%	95%	96%	
Marin et al. (2015)	0.87	12%	49%	–	84%	–	95%	
Giachetti, Ballerini & Trucco (2014)	0.879	13%	59%	82%	88%	92%	94%	
Cheng et al. (2013)	0.875	8%	51%	76%	86%	92%	–	
Aquino, Gegúndez-Arias & Marin (2010)	0.86	7%	46%	73%	84%	90%	93%	
ONHSD								
Our method (manual localization)	0.8927	10%	58%	86%	93%	97%	97%	
Our method (automatic localization)	0.8649	4%	43%	69%	87%	92%	96%	

Figure 8 Images from Messidor database.

(A–C) Examples of correct segmentations. (D–F) Examples of incorrect segmentations. The truth OD boundary is marked in green.

Figure 9 Images from ONHSD database.

(A–C) Examples of correct segmentations. (D–F) Examples of incorrect segmentations. The truth OD boundary is marked in green.

Discussion and Conclusion

According to Table 1, our results are similar to what has been found with other methods for OD localization. In the case of the Messidor dataset, the detection rate is slightly lower than what has been reported by other researchers. However, these higher rates do not translate necessarily into better segmentation performance as it can be seen in Table 2.

It is important to comment on the expected performance of the algorithms on the range of images found in routine clinical scenarios. Figures 6 and 7 show that the worst results correspond to low contrast images due to the illumination conditions and the presence of peripapillary atrophy. Illumination artifacts can also influence the results negatively, especially if they are located close to the OD. On the other hand, according to the conducted experiments, uneven illumination does not seem to be an issue, nor is it the presence of exudates which is overcome by taking advantage of the output of the Hough line detector as mentioned in the ‘Materials & Methods’. Choroidal vessels which are very pronounced could lead to wrong detection in localization procedures based on vessel density. The combination of vessel density with the vessel convergence information provided by the output of the Hough detector alleviates the problem. In general, a good performance can be expected as long as the images satisfy the hypothesis of the OD as a bright region with darker surroundings.

Regarding OD segmentation, Table 2 indicates that this problem is more difficult than localization. In Giachetti, Ballerini & Trucco (2014), some experiments were conducted to compare automatic and human performance using a subset of 300 images of the Messidor database and three experienced ophthalmologists. Intra and inter-operator average Jaccard coefficients of 0.932 and 0.92, respectively, were obtained. Therefore, it is clear that there is still some margin for improvement.

A simple circular model is proposed as the best solution for the OD segmentation problem. The potential theoretical performance of a circular approximation was obtained for each true OD contour by considering circles with varying radius and center coordinates at the center of mass of the corresponding ground truth mask, so that the ones which provided the best scores in terms of JC and MAD were selected. Average JC values of 0.940 and 0.932 were found for the Messidor and ONHSD databases, respectively, and average MAD values of 2.630 and 1.416 were obtained for the same datasets. The results in Tables 2 and 3 confirm that in a more realistic scenario, the circular model is still a very good choice. In what follows, we will only consider the segmentation method with automatic localization for the purpose of comparison. To the best of our knowledge, only the method proposed by Zilly, Buhmann & Mahapatra (2017), using ensemble learning based on CNN, outperforms ours in terms of the average JC on the Messidor dataset. However, their paper is mostly focused on the DRISHTI-GS dataset and hardly any detail is provided regarding the application of their method to Messidor. They claim that the OD is first localized by applying a CHT on the green channel but the obtained success rate is not reported neither is its impact on the results. Moreover, as the authors claim, one of the limitations of their algorithm comes from its non-deterministic behavior because of the sampling of points of the training images. The other methods report worse results for average JC and DC in both databases. With respect to the average MAD, our results are not the best but still satisfactory. This is something expected due to the lack of flexibility of the circular model to adapt to some of the true OD boundaries as compared to other models. The performance scores in Table 3 also show that our method is rather stable for values of JC ranging from 0.70 to 0.85 and provides, overall, desirable results across all intervals.

The fact that the segmentation method is performed on a subset of the original image reduces the number of artifacts and sources of error. Figures 8 and 9 show that poorer segmentations occur mainly due to low contrast between the inner and outer rings of the OD and, more importantly, because of the strong influence of the accuracy in the estimation of the disc center as it becomes clear from the results obtained. If the distance between the estimated OD center and the ground truth OD center is considerable, the proposed method will fail to produce an acceptable solution. Manual localization provides much better results. Therefore, future work will be focused on decreasing the localization error. Even in the case of perfect localization, there could be an issue with the segmentation of glaucomatous optic discs with high cup to disc ratio given that the maximum contrast could be achieved at the cup boundary according to (12). The confirmation of this point is also left for future investigation since it requires the use of retinal image databases intended for glaucoma diagnosis.

In conclusion, this research work seems to agree with the interesting discussion carried out by Aquino, Gegúndez-Arias & Marin (2010) about the advantages and disadvantages of using circular, elliptical and deformable models. They compared the circular approach with four different elliptical models and three deformable models and concluded that under appropriate background-contrast conditions, the deformable models render more accurate OD segmentations but under not so ideal conditions, a circular model is preferable because it provides a more robust and reliable solution. It is, therefore, another illustrative example of bias–variance trade-off in model fitting.

Supplemental Information

Supplemental Information 1 Code for automatic optic disc location and segmentation

Click here for additional data file.

Additional Information and Declarations

Competing Interests

Author Contributions

Data Availability

The authors declare there are no competing interests.

Jose Sigut conceived and designed the experiments, analyzed the data, reviewed drafts of the paper.

Omar Nunez conceived and designed the experiments, performed the experiments, analyzed the data, wrote the paper, prepared figures and/or tables, reviewed drafts of the paper.

Francisco Fumero and Rafael Arnay performed the experiments, analyzed the data, reviewed drafts of the paper.

Marta Gonzalez analyzed the data, contributed reagents/materials/analysis tools, reviewed drafts of the paper, provided advice about the medical aspects of this work.

The following information was supplied regarding data availability:

“Contrast based circular approximation for accurate and robust optic disc segmentation in retinal images”—Code. DOI 10.5281/zenodo.844911.

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
