# Peer review of "Contrast based circular approximation for accurate and robust optic disc segmentation in retinal images"

_PeerJ, doi:10.7717/peerj.3763_

## Round 0.1 · original submission · Major Revisions

In addition to the technical suggestions provided by the reviewers, please carefully review and edit the manuscript for linguistic issues.

Please comment on the expected performance of the algorithms on the range of images found in routine clinical scenarios (e.g. range of pigmentation/illumination where the disk may not be as bright relative to the background, the choroidal vessels are very pronounced etc).

Reviewer 1 ·

Basic reporting

Overall, the paper is written very professionally and reads well. With the exception of some rewording and some clarification here and there, the reporting of the work is very good. My main issue is with the figures. Figures 1 and 2 are a great opportunity to visualize the method, but aren’t very descriptive. The captions are too short, and only refer to the images by their abbreviations which require the reader to refer to the text to work out what they are. Consider expanding the captions to describe exactly what the figures are showing. This will help readers to understand the workflow more easily. A minor issue with Figure 3 is that the the arrows should be pointing towards the inner and outer rings, rather than their annotations. Figure 4 is quite confusing, mainly due to the use of colors. I would explicitly present them as four labelled semi-circles and annotate them with different colors.

The Results section demonstrates that the proposed method for OD localization has a very high success rate, comparing favorably with previous methods in Tables 1 and 2. The normalized Euclidean distance measure is reported, but isn’t very easy to interpret without examples. It would be helpful to annotate Figures 6 and 7 with the distance metrics calculated for these images. Tables 3 and 4 are very helpful in understanding how the JC overlap varies, especially in comparison with previous methods. It might be worth merging these tables into one, to avoid redundancy of reporting the Messidor results. The examples given in Figures 8 and 9 are also helpful to get a sense of how the results vary. Again, I would suggest annotating the images with the corresponding JC/DSC/MAD values for these images to get a sense of what a good/bad value looks like.

Here are some specific issues in the text that ought to be addressed:

- Lines 73/354: “In what respects to” is not grammatically correct. “With respect to” would be better.
- Line 105: I would suggest “partly inspired by” instead of “somehow inspired by”.
- Line 183: Is T8 the top-hat operator?
- Line 191: Please clarify how the Hough Transform is used to generate the image VC. Does the pixel value correspond to the number of Hough line intersections at each point?
- Line 232: What do you mean by a “more robust and efficient segmentation”?
- Line 242: I would say “thickness” of the ring, rather than width.
- Line 274: What is considered a “wrong” detection for the ONHSD dataset given that the localization accuracy was 100%?
- Line 341: Using “-“ to separate the JC and MAD values doesn’t read very well. Consider restructuring this sentence.
- Line 342: Not sure what you mean by theoretical performance, do you mean the best OD segmentation results when using manual OD localization (rather than auto)? Please clarify.
- Line 359-360: I would advise against using terms like “good” and “bad”. Consider phrasing like “desirable results across all intervals” and “Figures 8 and 9 show that poorer segmentations occur…”.

Experimental design

The paper describes a method for performing optic disc localization and segmentation from digital retinal images. The proposed method uses a series of morphological processing steps in combination with edge detection and a Hough transform to localize the OD center. OD segmentation is performed by finding a pair of concentric rings around the approximate disc center that maximizes the image contrast between the inner and outer ring. An alternative segmentation method using four semi-circles is also proposed.

The Introduction and Background give good context for the research question, and several existing methods for OD localization and segmentation are described. I would suggest that more detail be added on the limitations of these methods, as well as the contributions of the proposed method. The bias-variance trade-off is mentioned in the Conclusion, but should also be stated here. The authors do state that their method offers “much better results” than Aquino et al. but I would suggest restating this in less qualitative terms, perhaps referring to the higher DSC and localization accuracy for the Messidor dataset.

There are lots of parameters mentioned throughout the Methods section, which is great to see from a reproducibility standpoint. Although not explicitly stated, I assume that most (if not all) of them were derived experimentally. Can the authors confirm (a) which dataset(s) were used to determine these parameters, and (b) whether the method has been evaluated on an independent test dataset using these parameters? This will help to give an idea of how well the method is likely to generalize to other data.

Of the two methods presented (pure circular and four semi-circles), which was used to generate the reported results? Did you try both? This should be made clear.

Validity of the findings

The methodology is inspired by the work of Aquino et al. (2010), in which the authors argue that circular approximations of the OD are more reliable and robust than those that use elliptical or deformable models. The proposed method was evaluated against the publicly available Messidor and ONHSD datasets, yielding results that are comparable with (and in some instances, slightly outperform) previously reported methods.

In the Discussion, the authors state that their results fall in line with previous methods and refer to OD localization as a “solved problem”. However, I would advise against making such an assessment based on results from a single dataset. How well do these methods perform on other datasets? The authors also mention that a higher OD detection rate “does not translate into better segmentation performance”, referring to the segmentation results of Roychowdhury et al. However, it’s worth noting that in their paper, the authors compute the OD center from the segmentation, not the other way around. Hence, this dependency is reversed. Consider rewording this section.

The authors state that there is some margin for improvement in OD segmentation, referring to the intra- and inter-operator variability observed among ophthalmologists in a study by Giachetti et al. I would suggest that the authors take this opportunity to suggest how their method could be improved, and expand a little on what causes poor segmentation results.

The authors also discuss the limitations of a circular approximation, which gives a lower MAD than other methods. This makes sense given that optic discs rarely take the form of a perfect circle. What are the implications of this? Some discussion would be good, describing the situations in which robustness is more important than an accurately delineated boundary.

Additional comments

Overall, the paper is very well written and successfully demonstrates that a simple circular approximation of the optic disc can yield results that are comparable with a variety of previously reported approaches. The methods are described in excellent detail, and are evaluated by making appropriate use of several evaluation metrics. Furthermore, the MATLAB code is made available and can be readily tested against publicly available datasets.

Reviewer 2 ·

Basic reporting

- Line 166, I suggest to use a figure to illustrate the MAD metric.
- Line 173, please mentioned the method used for resizing.
- Captions on all figures: Could be better understood if you write commas between (a), (b), .... etc
- The section of OD localization is described in two steps 1) line 177 and 2) line 208. It could be a little more clear the description if you divide them in subsections.
- Figure 5, on the label of the vertical axis: normalized distance is the same as “error”? I suggest use “normalized distance” as in the text and in the caption.
- Lines 274, 278, 282, 314, 322. I suggest change good and bad, and right and wrong, by correct and incorrect.
- Line 287, “… described above”, explicitly mention the name or number of the section.
- Figures 8 and 9, list explicitly (a), (b), (c),…. etc.

Experimental design

- Lines 172 – 176. The set of parameters might work for the two image databases that you used because both were taken at 45 degrees. What happens if you have a set of images that are taken with other settings such as 50 or 60 degrees? With what criteria do you define the values of those parameters? It will not work well for any input set.
- Line 190 and 198, the threshold parameter does not depend on size, how was this parameter set? and does this value works the same for any image quality?
- Line 220 the x and y coordinates are the same as xc and yc of line 236?
- Line 220, what happened when there is more than one pixel with the maximum value? Which coordinates do you chose?
- Before OD segmentation, images were resized. It is not clear if for the segmentation process images kept resized or do you return to the original size (scale), for cropping windows.
- Line 237, what is the initial value for r. You increment the length of the radii and computed the differences of the averages how many times? How do you stop?
- Line 212, It is not clear how is defined Ic.

Validity of the findings

Lines 112-115. You did not really prove this with your results. It seems that with the use of manual centers extracted from the ground truth images you can get better segmentation results with your segmentation method. That only indirectly evaluates the method of OD location, as shown in table 1.
- Evaluation. The normalized Euclidean distance is defined as: D(x,y)=sqrt(sum[(xi-yi)^2/si]) where x and y are the extreme points where the distance is measured, and s = is de standard deviation. Why are you using R in equation 1?
- Evaluation. Lines 141-147. The most accurate and correct evaluation would be to calculate the distance between the centroids. That is, between the centroid of the ground true and the center obtained by your method. If your estimated center lies within the boundary of the corresponding ground truth is a very poor measure of error. You should do - Did you show your segmentation results using the first approach or the “alternative” one.
- Figure 5, If the distance is normalized range in vertical axis should be [0,1], why the range on Figure 5(a) is about [0,3] ? your evaluations between centers.
- From table 1, you argue that your results on OD detection were “similar” to those obtained by other authors. When in table 1 it is shown that approx. 63% (7 of 11 cases) are above yours in a range of differences of [0.5-0.75]. Line 328.
- On the other hand, from table 2, for OD segmentation results you argue that “.. other methods report worse results for average JC and DC in both databases” whose differences, range in the order of [0.001-0.007] for both JC and CD. However, for MAD, whose differences are about 2.00 for MESSIDOR you argue , Line 355, “.. our results are not the best but still satisfactory and similar…”
- The line dividing similar and worse, in the description of tables 1 and 2, is not very clear. And I could say that they are contradictory. For the OD location the differences are of tenths and you say similar, and for the OD segmentation the differences are of thousandths and you say they are worse than yours.
- If we start talking on the same comparative scales, I could conclude from tables 1 and 2 that the OD location had low performance but the OD segmentation is comparable with other authors. I suggest that you explain in more detail why your method is better than the others, if your performance is so similar.
- And as far as OD location is concerned, I think I should check how the other authors evaluate the location, because to evaluate whether or not the estimated center falls within the GT area, I do not think it is the general way in which other authors do this evaluation.

---

## Round 0.2 · accepted · Accept

Thanks for addressing the reviewer comments

Reviewer 1 ·

Basic reporting

All of my previous comments have been addressed.

Experimental design

All of my previous comments have been addressed.

Validity of the findings

All of my previous comments have been addressed.